# A Review of the Botany, Volatile Composition, Biochemical and Molecular Aspects, and Traditional Uses of *Laurus nobilis*

**DOI:** 10.3390/plants11091209

**Published:** 2022-04-29

**Authors:** Antonello Paparella, Bhagwat Nawade, Liora Shaltiel-Harpaz, Mwafaq Ibdah

**Affiliations:** 1Faculty of Bioscience and Technology for Food, Agriculture and Environment, University of Teramo, Balzarini, 1, 64100 Teramo, Italy; apaparella@unite.it; 2Newe Yaar Research Center, Agricultural Research Organization, Ramat Yishay 30095, Israel; 26bhagwat@gmail.com; 3Migal Galilee Research Institute, Kiryat Shmona 11016, Israel; liora@migal.org.il; 4Environmental Sciences Department, Tel Hai College, Upper Galilee 12210, Israel

**Keywords:** bay, biochemical, *Laurus nobilis*, traditional uses, essential oil

## Abstract

*Laurus nobilis* L. is an aromatic medicinal plant widely cultivated in many world regions. *L. nobilis* has been increasingly acknowledged over the years as it provides an essential contribution to the food and pharmaceutical industries and cultural integrity. The commercial value of this species derives from its essential oil, whose application might be extended to various industries. The chemical composition of the essential oil depends on environmental conditions, location, and season during which the plants are collected, drying methods, extraction, and analytical conditions. The characterization and chemotyping of *L. nobilis* essential oil are extremely important because the changes in composition can affect biological activities. Several aspects of the plant’s secondary metabolism, particularly volatile production in *L. nobilis*, are still unknown. However, understanding the molecular basis of flavor and aroma production is not an easy task to accomplish. Nevertheless, the time-limited efforts for conservation and the unavailability of knowledge about genetic diversity are probably the major reasons for the lack of breeding programs in *L. nobilis*. The present review gathers the scientific evidence on the research carried out on *Laurus nobilis* L., considering its cultivation, volatile composition, biochemical and molecular aspects, and antioxidant and antimicrobial activities.

## 1. Introduction

*Laurus nobilis* L. is an aromatic and medicinal plant belonging to the Lauraceae family, which comprises approximately 2500–3500 species [1]. The genus *Laurus* consists of two species: *Laurus azorica* and *Laurus nobilis*. The generic half of the binomial, *Laurus*, comes directly from the Latin name for the tree and is probably taken from a more ancient Celtic word, *blaur*, meaning green, while *nobilis* is a Latin word meaning noble and famous [2]. It is an evergreen shrub native to Mediterranean regions, also known as sweet bay, bay laurel, Grecian laurel, true bay, or simply bay [3,4]. Different ploidy levels (2n = 36, 42, 48, 54, 60, 66, 72) have been reported in *Laurus* [5], with tetraploidy (2n = 4x = 48) being the most frequent karyotype. Thus, the chromosome number in most of the laurel is 2n = 48. Out of 48, 40 are metacentric, and eight are submetacentric chromosomes [6]. The mean chromosome length is 4.01 ± 0.1 µm [7].

Turkey is the major producer of *L. nobilis* and exports it to 64 countries [4,8]. Almost 97% of the world’s total production comes from Turkey [9]. The amount of annual production ranges between 7000 and 7500 tons [10].

*L. nobilis* is not only an aromatic plant but has also been valued for thousands of years for its cleansing properties. *L. nobilis* is widely cultivated in many regions of the world, primarily used as a culinary herb [4]. The different body parts of *L. nobilis* and its essential oil (EO) have been recognized to possess many interesting properties that have potential applications in many areas, including agriculture, medical, food, pharmaceutical industries, etc. The leaves are commonly used as a spicy, aromatic flavoring agent for soups, fish, meats, stews, puddings, vinegar, and beverages. The pharmaceutical properties of *L. nobilis* leaves and fruits have been known since Dioscorides time [11]. Due to its antimicrobial and insecticidal activities, bay is used in the food industry as a food preservative [12,13]. The cosmetic industry also uses the *L. nobilis* EO in creams, perfumes, and soaps [14,15]. Essential oil exhibits beneficial functions such as antibacterial, antifungal, and antioxidant activities [13,16]. The seeds are reported to have antiulcer and antidiabetic effects [17]. Currently, the request for natural products for novel applications is increasing day by day; for example, bay laurel berries are used as a natural anthocyanin instead of synthetic dyes in the food, pharmaceutical, and cosmetic industries [18].

The commercial value of this species derives from its essential oil and from its volatiles in general. The volatiles are secondary plant metabolites found in different parts of plants, including flowers, roots, bark, leaves, seeds, fruit, and wood, produced in the cytoplasm and plant cells plastids [19]. The volatiles are odorous compounds (<C15) with low molecular mass (<300 Da), high vapor pressure, low boiling point, and a lipophilic moiety [20]. Low molecular terpenes are easily volatilized at room temperature and represent significant floral scents and essential oils of herbs, vegetables, and fruits [21,22]. Essential oils are volatile, complex mixtures of compounds characterized by a strong flavor and aroma formed by plants as specialized metabolites. Volatile compounds are essential components of flavor and aroma in many crops. It has recently been suggested that plant volatiles, due to their aroma, provide sensory clues as to foodstuffs’ health and nutritional status [23,24,25].

This review aims to provide an overview of the research carried out on *L. nobilis* so far, considering its cultivation, geographical distribution, EO composition, genetic and molecular aspects, as well as antioxidant and antimicrobial activity.

## 2. Botany and Distribution

*L. nobilis* is an evergreen shrub that can reach a height of 15 to 20 m in the natural environment. However, the dimensions are usually smaller (4–6 m) in gardens and yard spaces. It can be grown as a single-trunked tree or a multi-trunked shrub and adapts well to pruning and shaping, and can be used for topiary or grown as a standard. The bark is smooth and has an olive green or reddish hue. The leaves are lanceolate or lanceolate-acuminate, having an alternate leaf arrangement with short stalks [26]. They are 5–8 cm in length or longer and 3–4 cm wide, coriaceous, pellucid-punctate, and with revolute, entire wavy margins. The leaves upper surface is glabrous and shiny, olive-green to brown, while the lower surface is dull olive to brown with a prominent mid-rib and veins [26,27]. When crushed, the leaf has a characteristic fragrance, and its taste is bitter and aromatic [26]. The plants are dioecious, with star-shaped male and female flowers on different plants [4]. The flowers are small, yellow-white with four tepals. It blooms in the spring, between March and May. The flowers are fragrant, gathered in inflorescences that develop in leaf axils or branch tips. The female inflorescence possesses few flowers with a superior ovary containing one loculus, while the male inflorescence has numerous flowers with several stamens attached to the corolla. The fruits are berry type, single-seeded with a loose kernel. The olive-like berries of bay laurel are green in color. Firstly, when it matures, it becomes a bright bluish-black color. The *L. nobilis* dried fruits are drupaceous, ovoid, about 15 mm long, and 10 mm wide [6].

## 3. Origin, Domestication, and Spread

*L. nobilis* is the true laurel of Greek and Roman mythology. According to mythology, this plant was born from the metamorphosis of Daphnes, a nymph who wanted to escape from Apollo. When Daphne, daughter of the earth goddess Gaia, was pursued by Apollo, the slayer of her bridegroom, she prayed to the Gods to help her, and they transformed her into a laurel tree to avoid the lustful pursuit of Apollo. After that, Apollo crowned himself with a wreath of laurel leaves to show his love for Daphne and declared the tree sacred to his divinity [16]. Laurel has always symbolized victory and merit, and a baccalaureate (*baca lauri*, Latin for “laurel berry”) is still a symbol of accomplishment. Since ancient times, the plant has been known and associated with several myths and is considered a symbol of victory, glory, wisdom, and honor. Even today, its leaves are used to make crowns for university graduation ceremonies as a symbol of honor and great recognition.

This plant is widespread in the Mediterranean countries, e.g., Algeria, Turkey, Spain, Morocco, Italy, Greece, and Portugal, and cultivated in other temperate and warm parts of the world (Figure 1) [4,24]. It is also found in tropical and subtropical Asia, Australia, the Pacific, and South Asia. Turkey, Italy, Belgium, Algeria, France, Tunisia, Iran, Morocco, Serbia, Greece, Portugal, Centers America, and the Southern United States are the commercial production centers of bay leaves [4,24]. It is a slow-growing, natural evergreen member of Mediterranean region vegetation. More than 20 million years ago, laurel forests (Laurisilva) covered a large part of the Mediterranean basin in the Tertiary period. However, due to glaciations in the Quaternary period, these subtropical forests moved to more temperate areas: North Africa and the Macaronesian archipelagos [28]. It is grown commercially for its aromatic leaves but is also widely cultivated in Europe and the USA as an ornamental plant [29].

## 4. Cultivation and Conservation

### 4.1. Growth Conditions

*L. nobilis* shrub grows in areas with a mild, cold climate associated with areas near the sea, ravines, and humid, shady valleys. It is not found at high altitudes because it suffers from frosts. It can grow in the garden also as a hedge. The optimal conditions for *L. nobilis* growth are sunny and well-drained soils. *L. nobilis* shrub grows best in areas where annual daytime temperatures range from 17 to 25 °C but can tolerate 8–30 °C (Tropical Plants Database). The shrub can survive at temperatures of about −5 °C with occasional short-lived lows to −15 °C when dormant, but young growth can be severely damaged at −1 °C. *L. nobilis* shrub requires a mean annual rainfall of 600–1000 mm, but tolerates 300–2200 mm [30]. *L. nobilis* prefers a position in full sun but succeeds in light shade and even in fairly dense shade. Any soil with moderate fertility is suitable for its growth, but this species prefers moisture-retentive, well-drained fertile soil [31]. The overabundance of water leads to root rot. The bay plant is wind-hardy, but it does not like extreme maritime exposure and cold, dry winds [32]. It prefers a pH of 5–6.5 but can tolerate 4.5–8.2 [31].

### 4.2. Propagation

*L. nobilis* can be propagated by seeds, stem cuttings, micropropagation, and in vitro culture techniques [33]. Bay seeds are dormant, but the type of dormancy is unknown yet [34]. The L. nobilis seeds are covered with a fleshy pericarp and a hard seed coat. The endosperm occupies most of the seed. Pericarp, seed coat, and perhaps inhibitor(s) associated with the coat are reported to be responsible for seed dormancy. Removing the pericarp alone cannot break dormancy, and the seed coat needs to be removed to have a high germination percentage without stratification. Seed dormancy is broken by soaking the seeds in water under room conditions for approximately ten days to remove the pericarp and stratify the seeds at 4 ± 1 °C for 30–45 days [34,35].

Further, Konstantinidou et al. [36] demonstrated the recalcitrant nature of bay seeds and reported that the most effective way of conserving the seeds is moist storage at 0 ± 1 °C for four months without previous drying [37]. Under warm climatic conditions, seeds may take several months to a year to germinate. Most bay trees are commercially prepared by taking semi-ripe cuttings in late summer [38]. Beginning to harden wood makes the best cuttings, but even these take up to three months to root under the best conditions. An additional propagating method is by layering the *L. nobulis* plant. This method is often successful but slower than cuttings and requires extensive gardening skills [37]. These propagation techniques using cutting, seeds, and layering are very slow and do not ensure homogeneity [39]. In consequence, this species has become about to be endangered due to overcutting and non-efficient conservation.

The limitations of traditional propagation techniques, such as poor fruit sets and a very low germination rate, prompted researchers to develop different tissue culture techniques for propagation. Souayah et al. [33] used micropropagation by axillary buds of mature plants, which revealed significant rooting differences due to the type of cutting. The buds were found to affect the induction of the first root. Shoot multiplication and elongation were obtained using benzyl-aminopurine combined with gibberellic acid, while root induction was obtained in MS medium salts at 1/3 strength with naphthalene-acetic acid [33]. Germination showed that MS basal medium was more efficient than half MS medium and woody plant medium (WPM), and this result was confirmed by the measurement of the aerial parts length, the roots and the number of axillary buds for nine weeks [40]. Somatic embryogenesis was induced in embryo culture on half MS medium supplemented with NAA (8 mg/l) to promote callus induction and somatic embryogenesis [39]. Further, the same authors reported survival of 75% of plantlets derived from the callus in the greenhouse; and phenotypically average growth of regenerated plants.

### 4.3. Pests and Diseases

*L. nobilis* is susceptible to pests like aphids (Aphidoidea), scale insects (Pseudococcidae), jumping plant lice (Psyllidae), thrips (Thysanoptera), caterpillars of codling moth (*Cydia pomonella*), carnation leafroller (*Cacoecimorpha pronubana*), and mites [41,42]. The most common disease of *L. nobilis* is leaf spot [43], but it also suffers from sooty molds (*Cladosporium* sp., *Aureobasidium* sp., *Antennariella* sp., *Limacinula* sp., *Scorias* sp., and *Capnodium* sp.) [44].

### 4.4. Harvesting

Bay leaves can be harvested during the entire year because the plant is evergreen. However, leaves are generally collected for the herbal plants when the plants bear flowers [6,12,38]. The berries are collected at about 40% moisture when they reach physiological maturity. Usually, one or two harvests a year are recommended for the highest yield and the highest dry leaf quality. Weather conditions like dew, high humidity, and rains are avoided during harvesting as these can cause deterioration and discoloration [37]. Collection of the leaves is generally carried out by hand or using small farming tools such as rakes. Sometimes, the plant stems are cut, and the leaves or fruits are removed following the harvesting [45]. Bay leaves are classified according to shape, size, color, and aroma before packaging [46]. According to the various quality standards and consumer preferences, the leaves are packaged and kept in a cool and dry place. The suggested storage conditions for spices are 10–15 °C and 55–65% relative humidity (RH) [47].

## 5. Plant Genetic Resources

This plant has been extensively studied for its EO and its volatile composition, having various medicinal properties. However, almost no effort has been made regarding the breeding aspects such as genetic resource characterization, conservation, genetic improvement, and genomics. Variations in *L. nobilis* have not been fully cataloged. Usually, the leaves are collected from naturally grown trees, which have a wide variation in morphological and biochemical traits. For example, there are no registered *L. nobilis* cultivars in Turkey [48]. Recently, Nadeem et al. [48] described *L. nobilis* orchards with around 600 genotypes collected from 300 different geographical locations in Turkey’s Marmara, Aegean, and Mediterranean regions. A collection of 203 genotypes were characterized for morphological and biochemical traits from Hatay province, and 95 genotypes were selected for their superior characteristics [49]. In Figure 2 and Table 1, a list of *L. nobilis* cultivars described in recent publications and web sources is reported.

According to gardeners’ preference, most of these cultivars are developed by commercial nurseries, like “Sunspot,” a cultivar with gold-variegated foliage, and “Brilliant times,” with a reddish stem with bright yellow leaves. There is an urgent need to develop the *L. nobilis* genebank for the collection, documentation, regeneration, distribution, and conservation of genotypes worldwide. The germplasm must be collected from diversity-rich areas of the world to enrich the genetic resources. It will act as a reservoir of useful genes and alleles, contributing to the genetic enhancement and providing raw material for further improvement programs. Several studies have been conducted on the *L. nobilis* species and their variability, especially for the EO and its volatile composition and various medicinal properties. These collections could be further utilized for *L. nobilis* improvement.

## 6. Characterization and Evaluation

### 6.1. Characterization for Essential Features and Classification

A total of 203 genotypes were characterized by various morphological and biochemical traits from Hatay province, Turkey, and 95 genotypes were selected for their superior traits [49,50], considering the following characteristics: fruit weight, kernel weight, kernel ratio, dry leaf ratio, leaf area, berry oil content, berry flesh oil content, kernel oil content, ovality coefficient, lauric acid ratio, oleic acid ratio, palmitic acid ratio, chlorophyll SPAD value, EO content, 1,8-cineol content, EO components [50]. Furthermore, 149 female trees were preselected from the same region, and then four berries of 48 female genotypes were characterized for their pomological and chemical properties. Significant variation was recorded among the genotypes for different traits, including berry weight (0.77–1.76 g), kernel weight (0.49–1.12 g), kernel ratio (51.73–77.44%), dry matter ratio (44.89–69.44%), berry oil ratio (18.92–37.85%), berry flesh oil ratio (20.76–53.98%) and kernel oil ratio (11.75–27.49%). In the same study, the fatty acid content ranged between 12.74–31.19% for lauric acid, 12.35–19.91% for palmitic acid, 30.35–44.43% for ole acid, and 15.93–26.75% for linoleic acid. Among the genotypes, genotype K9 for high lauric acid and low palmitic acid ratio, genotype ER6 for berry weight, B30 for kernel weight, and ER14 for kernel oil ratio were found to be promising genotypes.

Characterization of wild *L. nobilis* trees from Israel revealed twenty-one distinct varieties, and this inventory contains trees having dwarfs with tiny leaves, huge bays with a heavy concentration of dark green leaves, and scents that ranged from an EO described as having a “good lemony” aroma to others with almost no odor at all [2]. Recently, 12 distinct populations comprising 1200 plant samples of *L. nobilis* in the Croatian Adriatic area were characterized for EO content in leaves, and on this basis, it was concluded that the populations of bay laurel from south-east Croatian Adriatic islands and coastal areas accumulate a higher quantity of EO in comparison with the populations of north-west islands and coastal areas [51].

### 6.2. Characterization of Laurus nobilis for its Essential Oil and Volatile Composition

#### 6.2.1. Leaf

The chemical composition of *L. nobilis* leaves from different origins has been reported to consist of 1,8-cineole as the significant volatile in all the cases (Figure 3A). Apart from this, sabinene, *α*-terpinyl acetate, linalool, *α*-pinene, *α*-terpineol, and methyl eugenol are among the major volatiles reported from the EO with varying concentrations from different locations (Table 2). The most abundant leaf EO constituents 1,8-cineole, *α*-terpinyl acetate, linalool, and sabinene were comparable to the respective values reported from different countries like Albania, Argentina, Bulgaria, Iran, Turkey, and Serbia (Table 2). In particular, leaf EO from India and Nepal was found to have linalool as a significant component, followed by 1,8-cineole and *α*-pinene [52]. The 1,8-cineole concentration in the leaf EO of *L. nobilis* from Bulgaria was similar to previous reports. The 1,8-cineole performs essential ecological functions, such as repelling insects and deterring herbivores [53,54]. The bicyclic monoterpenes *α*-pinene and *β*-pinene, among the frequently occurring volatiles in bay leaves, are lipophilic, insecticidal, sedative, fungicidal, and anticarcinogenic effects [55]. The phenylpropene derivatives eugenol, methyl eugenol, and elemicin are also reported in the bay leaf, and these are responsible for the spicy aroma of the leaves and are significant factors determining its sensory quality. Eugenol and methyl eugenol have anesthetic, hypothermic, muscle-relaxant, anticonvulsant, and anti-stress activities on humans and antifungal, antibacterial, antinematodal, or toxicant roles against pathogens and insect herbivores [56].

In a study carried out in Turkey, the composition of volatile organic compounds (VOCs) from leaves of young and old shoots of *L. nobilis* were significantly different. The leaves of young shoots contained higher amounts of *β*-pinene, *α*-pinene, linalool, *α*-terpineol, 2-hydroxy−1,8-cineole, and some sesquiterpenes, while 1,8-cineole, sabinene, sabinene hydrates, terpinene−4-ol, *α*-terpinyl acetate, eugenol, and eugenol methyl ether were found in higher concentrations in the leaves of old shoots [12]. Yahyaa et al. [90] (2015) found that the leaf stage and the gender of the plant significantly affected leaf volatile composition. Organs of female plants generally contained more terpenes than the corresponding male organs. In general, all-male plant parts had a consistently lower concentration of 1,8-cineole than female plants. In contrast, the leaves and flowers from male plants had considerably more *δ*-elemene than the corresponding female organs. Apart from these factors, the environmental conditions, location, and season during which the plants are collected and dried, extraction, and analytical conditions contribute to the differences in major volatiles composition.

#### 6.2.2. Fruit

The fruits of this dioecious plant species are olive-like black berries. According to Marzouki et al. [87], the VOCs from different plant organs contain similar compounds, but the quantitative differences between all main compounds are quite large. In this study, volatiles such as 1,8-cineole, sabinene *α*-terpinyl acetate, methyl eugenol, eugenol, and linalool were the main components of the EO of leaves, buds, and flowers, but the same compounds were present in small quantities in the fruits [87]. Kilic et al. [12] reported (*E*)-*β*-ocimene as a significant fruit volatile that is not present in Turkish bay leaves, while Hafizoǧlu et al. [91] found 4-terpineol to be the main component in the fruit EO from Turkey (Figure 3B).

Castilho et al. [92] reported (*E*)-*β*-ocimene and germacrene D in Portuguese *L. nobilis*, while (*E*)-*β*-ocimene in Tunisian bay was found as a predominating fruit volatile [93]. Yahyaa et al. [90] analyzed green and black fruit separately and found (*E*)-*β*-ocimene, *γ*-murolene, (*E)-α*-farnesene, *γ*-cadinene, and *δ*-cadinene in green fruits. Conversely, 1,8–cineole, (*E*)-*β*-ocimene, (*E*)*-α*-farnesene, *γ*-cardinene, and *δ*-cadinene were abundant in black fruits. It was also remarkable that the fruits on the female plants contained high levels of two sesquiterpenes, *γ*-cadinene, and *δ*-cadinene, that were found in much lower levels elsewhere in the plant. Fruits displayed the most divergent volatile profile from all other organs. The norisoprenoid volatiles such as 6-methyl-5-hepten-2-one (MHO), pseudoionone, and *β*-ionone were present only in the fresh pericarp of the black fruits. These norisoprenoid volatiles are distributed in numerous fruits and considered important and potent flavor, aroma, and scent contributors in many fruits and fruit-based foods due to extremely low odor thresholds.

#### 6.2.3. Root

*L. nobilis* roots’ volatile composition was also determined by Yahyaa et al. [90] by auto-headspace-solid-phase microextraction-GC-MS (SPME-GC-MS) analysis of both male and female roots. As shown in Figure 3C, the main volatiles in the roots were: 1,8-cineole *α*-terpinyl acetate, terpinene-4-ol, *p*-cymene, and *δ*-cadinene [90]. However, most of the volatiles were obtained in higher concentrations in female roots.

#### 6.2.4. Flower

The major study conducted to ascertain differences in the volatiles emitted from the whole living male and female flowers or different parts, found that the volatile fraction of the whole flowers were dominated by (*E*)-ocimene, which constituted 65.3% of the female flowers and 45.7% of the male flowers, followed by 1,8-cineole constituting 20.5% of female and 26.3% of male flowers [88]. Among the pollen volatiles, limonene (11.6%) was the principal compound, followed by 1,8-cineole (9.1%), terpinolene (4.7%), and γ-terpinene (4.5%). The significant differences between the VOCs of staminoids and whole female flowers were: (*E*)-ocimene (17.8% vs. 65.3%, respectively), *β*-caryophyllene (15.4% vs. 0.4%), (*Z*,*E*)-*α*-farnesene (10.3% vs. 0.5%), 1,8-cineole (7.9% vs. 20.5%), *β*-elemene (5.3% vs. 0.1%), and germacrene D (5.1% vs. 0.1%). The monoterpene hydrocarbons decreased while oxygenated monoterpenes and sesquiterpenes were increased in pollens and staminoids [88]. There were significant differences in volatiles’ composition between different organs and male and female Israeli bay plants. The biggest gender differences were observed in the flowers, where female flowers contained predominantly monoterpenes as well as eugenol and methyleugenol, whereas male flowers contained mostly sesquiterpenes and benzaldehyde. These differences could represent an olfactive gradient that could guide pollinators to the food rewards, thus acting analogously to the visive nectar guides.

## 7. Factors Affecting the Volatile Composition in *L. nobilis*

### 7.1. Extraction Methods

The market demand for *L. nobilis* EO and volatiles have increased remarkably due to their intriguing biological activities and wide applications. Therefore, new extraction methods were proposed to improve recovery without altering the qualitative features. Traditionally, these metabolites are extracted from the plant matrix by hydrodistillation (HD) and steam distillation (SD), but these methods can have disadvantages. In particular, extensive hydrolysis and thermal degradation can cause a characteristic off odor. During HD. On the other hand, SE gives EOs with a higher content of waxes and/or other high molecular mass compounds, often characterized by a concentrated scent, which is very similar to that of the material from which it was derived. A further drawback of SE is that small amounts of organic solvents can pollute the extraction product [84]. Recently, alternative methods have been developed to decrease the environmental impact by less or no solvent and less energy, maintaining a good yield in volatiles (Table 3).

Among these improved techniques, supercritical fluid extraction (SFE) is a separation technique in which the yield and selectivity can be controlled to some extent by changing the pressure and temperature of the fluid [82,84]. These studies demonstrated that the lighter compounds like hydrocarbon and oxygenated monoterpenes were extracted in shorter times than the heavier hydrocarbon and oxygenated sesquiterpenes. They also found a decrease in monoterpenes, their oxygenated derivates, and diterpenes. Ozek et al. [95] reported a decrease in monoterpene hydrocarbon content in supercritical CO_2_ extracts (6.8–5.1%) in comparison to the hydro–(19.5%) and steam (16.5%) distilled EOs. Caredda et al. [74] observed remarkable differences in 1,8-cineole and methyl eugenol contents after the first and fourth hour of extraction (1,8-cineole, 30% vs. 2% methyleugenol, 6.8% vs. 16.4%). A higher amount of sesquiterpenes were obtained when extraction was allowed to go beyond 90 min, while the monoterpene hydrocarbon extraction was almost completed at this point. Therefore, extraction time can significantly affect the composition of *L. nobilis* EO [93].

Microwave extraction is another solvent-free extraction method employed for volatiles estimation in *L. nobilis*. Compared with the conventional method, the oxygenated compounds are extracted in higher amounts while monoterpenes are decreased (Table 3). Bendjersi et al. [96] found some VOCs such as tricyclene, decanal, aceteugenol, and germacrene-D-4-ol only when using a microwave oven for extraction, *p*-cymene, *trans-β*-ocimene, pinocarvone, myrtenol, *trans*-carveol, carvone, and *β*-selinene were detected only in HD. Similarly, Flamini et al. [11] also detected *β*-elemene, spathulenol, and epi-*γ*-eudesmol in HD fraction, and *δ*-terpineol and borneol using microwave extraction.

Finally, Díaz-Maroto et. [81] developed a Simultaneous Distillation Extraction (SDE) method, where volatile levels are less concentrated. SDE showed higher extraction yields and better coefficient of variation values. The extracts obtained by SPME were rich in oxygenated terpenes (95.7%) compared with those in SDE (83.4%), while hydrocarbon monoterpenes were considerably lower (3.6%) than SDE (15.7%) [81] (Table 3). Moreover, Boulila et al. [94] used for the first time hydrolytic enzymes, viz. cellulase, hemicellulase, xylanase, for bay volatiles extraction, and they found an increase in 1,8-cineole, methyl eugenol, terpinen-4-ol, *α*-terpineol, and caryophyllene oxide, compared with the conventional extraction method. The enzyme pretreatment did not induce any transformation of the volatile components, but it contributed presumably to the liberation of some glycosidically bound volatiles, which increased their amounts in the EO.

### 7.2. Drying Methods

The volatile composition can also be affected by different drying methods. In this regard, different researchers conducted experiments to trace *L. nobilis* volatiles variations. Díaz-Maroto et al. [81] reported substantial losses in volatiles in frozen and freeze-dried bay leaves, except eugenol, elemicin, spathulenol *β*-eudesmol, whose concentration levels increased. The concentrations of certain oxygenated terpenes, such as 1,8-cineole, linalool, and geraniol, decreased slightly after air or oven drying (45 °C), but in the case of the *α*-terpinyl acetate, this decrease was more pronounced, i.e., 45%. Similarly, the lowest amounts of *α*-terpinyl acetate (8.6%) and *γ*-terpineol (1.2%) were obtained at 40 °C in the oven by Hadjibagher Kandi and Sefidkon [69]. In another study, Sellami et al. [98] found sesquiterpene hydrocarbons, particularly in air-dried leaves (0.25–11.45%), whereas oxygenated sesquiterpenes were especially detected in fresh leaves (0.08–0.43%). The effect of air and oven drying (45 °C and 65 °C) resulted in the loss of most of the monoterpene hydrocarbons, and this loss was more pronounced with a temperature increase from 45 to 65 °C in oven drying. Along with the disappearance of some volatiles, these authors also observed specific volatiles in air and infrared (IR) (65 °C) drying: tricyclene, trans-2-hexenal, trans-2-hexenol, α-terpinene, terpinolene, camphor, linalyl acetate, allylanisole, myrtenyl acetate, α-terpineol, borneol, valencene, geranyl acetate, myrtenol, nonadecane, and spathulenol. However, these compounds were absent in the fresh leaf extract. Due to drying, moisture moves by diffusion to the leaf surfaces and drags EO with it and these monoterpenes might have more affinity to the water fraction or most of bay laurel monoterpene hydrocarbons may be stored on or near the leaf surface and thereby, they were lost with water during drying process [98,99].

### 7.3. Season

The sample collection season was also found to be responsible for some changes in the volatile composition of bay leaves. Kilic et al. [12] reported a higher concentration of odor contributing compounds in autumn (e.g., linalool, 2–4-fold; eugenol, 4–10-fold) compared to July harvested leaf samples, leading to a better flavoring quality in the autumn samples despite their lower EO content. The leaves harvested in July were up to 6–8-fold richer in (*E*)-isoeugenol compared to the samples harvested in October [12]. Analysis of Iranian bay leaves from March, June, September, and December months revealed that the June sample had a higher content of 1,8-cineole (40.25%) than others. During these months, *δ*-3-carene, camphor, camphene, and sabinene were reported as minor compounds. No seasonal changes were found in the concentration of eugenol, methyl eugenol, and *α*-terpenyl acetate [72].

Therefore, it is of paramount importance for every location to characterize bay leaf volatiles for qualitative and quantitative changes according to the season, considering that these changes can affect biological activities.

## 8. Molecular Characterization

Almost no efforts have been made to preserve the genetic resources of *L. nobilis*. In fact, to the best of our knowledge, no germplasm collections affected the breeding programs of this plant species. Overall, scanty efforts have been directed towards laurel improvement, and most of the work has been focused on the EO, its volatile composition, and its various medicinal properties.

### 8.1. Molecular Markers

Genetic diversity among individuals and/or populations can be determined using morphological, biochemical, and DNA-based molecular markers. There have been relatively few reports of molecular marker-based approaches to bay improvement, and not even the genetic map is available for this important plant. The amplified fragment length polymorphism (AFLP) markers were utilized to investigate genetic relationships between 14 populations of *L. nobilis* and *L. azorica* from different geographical areas, including the Iberian peninsula, the Canary and Madeira Islands, France, and Italy [100]. The AFLP analysis of these samples revealed a low genetic similarity between the Iberian populations, including populations from Northern Spain, and the rest of the populations analyzed from France and Italy. The Iberian peninsula accessions also displayed higher genetic similarity to accessions from the Canary Islands and Madeira, originally identified as *L. azorica*, than to samples from the Mediterranean area, morphologically classified as *L. nobilis* [100]. The plastid DNA (cpDNA) sequence (trnK-matK, trnD-trnT) analysis of 57 *Laurus* populations and three Lauraceae genera revealed monophyly for *Laurus* with low sequence variability within *Laurus*. Three lineages were obtained: a first clade containing eastern lineage, corresponding to Turkey and the near East populations, a second clade in the Aegean region, and lastly, a western clade that grouped all populations from Macaronesia and the central and western Mediterranean regions [29].

The inter-simple sequence repeats (ISSRs) were also used for genetic analysis of *Laurus* species, including eight taxa (representing three tribes, four genera, and seven species) grown in Egypt [101] and *L. nobilis* populations distributed in the Aegean region [102]. The ISSR markers were sufficient to resolve the relationship within Lauraceae [101], while the polymorphism rate for these markers is slightly low among *L. nobilis* populations in the Aegean/Turkey region [102]. High-resolution markers allow identification of many genotypes that cannot be detected using conventional markers. Microsatellites/simple sequence repeats (SSRs) are sequences of one–to six-nucleotide motifs repeated in tandem, which can be used to characterize individuals at a higher level of resolution. These highly polymorphic, multi-allelic, and codominant markers are considered ideal markers for population and diversity studies [103]. Genetic analysis of sixty-six Mediterranean laurel trees separated *L. nobilis* into two main gene pools, one from western (Tunisia, Algeria, and France) and the other from eastern Mediterranean (Turkey) [104]. The same authors also reported that *L. nobilis* has a higher genetic differentiation than the other angiosperms. Moreover, Arroyo et al. [105] characterized a total of 63 genotypes containing 26 from Macaronesian islands (*L*. *azorica*) and 37 from the Mediterranean SEA (*L. nobilis),* with 20 newly designated polymorphic SSR markers. One hundred ninety-six alleles belonged to the *L. nobilis* species, and 222 alleles belonGED to the *L. azorica* species. Recently, 95 bay laurel genotypes selected from the flora of Hatay province for their superior characteristics were characterized using six SSR markers, and a total of 82 alleles were obtained with a mean of 16.4 of five polymorphic loci [50].

DNA advancement based on different molecular markers and molecular breeding offered many new strategies to breeders to resolve the problems encountered during conventional breeding. In recent years, molecular tools have been used to elucidate some aspects of genetic diversity in aromatic species, the genetic relationships between different cultivars, as well as the comparison between molecular marker analysis and plant chemical composition [106]. However, in the case of *L. nobilis*, utilization of these technologies is at a very early stage, and in the future, more concentrated efforts are needed in this direction.

### 8.2. Identification and Characterization of Novel Genes

With their abundant monoterpenes and sesquiterpenes, Bay leaves are used to impart flavor and aroma to foods. To identify terpene synthases (TPSs) involved in producing these volatile terpenes, Yahyaa et al. [90] performed RNA sequencing to profile the transcriptome of *L. nobilis* leaves. Bioinformatics analysis led to the identification of eight TPS complementary DNAs. The characterization of the enzymes encoded by three of these complementary DNAs revealed the following enzymes: a monoterpene synthase belonging to the TPS-b clade, catalyzing the formation of mostly 1,8-cineole; a sesquiterpene synthase belonging to the TPS-a clade, catalyzing the formation of mainly cadinenes; and a diterpene synthase of the TPS-e/f clade, catalyzing the formation of geranyllinalool [90]. Furthermore, a full-length cDNA encoding a carotenoid cleavage dioxygenase (*LnCCD1*) was isolated from *L. nobilis* fruits and overexpressed in *Escherichia coli* heterologous system. The recombinant protein was able to cleave a variety of carotenoids at the 9,10 (9′,10′) and 5,6 (5′,6′) positions to produce 6-methyl-5-hepten-2-one, pseudoionone, *β*-ionone, and *α*-ionone. These results suggest a role for LnCCD1 in the norisoprenoid biosynthesis because the volatile norisoprenoids 6-methyl-5-hepten-2-one, pseudoionone, and *β*-ionone accumulated in *L. nobilis* fruits in a pattern reflecting their carotenoid content.

### 8.3. Sex Determination

Considering that *L. nobilis* is a dioecious plant, identifying sex is the most daunting task to differentiate male and female flowers at an early stage of development. Morphological characters of flower height, flowering time, and flower numbers were used for the sex differentiation in *L. nobilis*. Male flowers were reported to have 8–14 stamens, while female flowers have 2–4 staminodes. Male plants produce a higher number of flowers than female plants, and male flower life is shorter than female flowers [48,107,108]. Mature male flower height is between 5.7–6.2 mm, which is more or less double the size of the female flower [109]. Royandazagh [110] used flow cytometry to determine the sex of *L. nobilis* at an early stage by analyzing the nuclear DNA contents of 58 seedlings and 50 samples from known male and female plants. The study findings showed that the male DNA content is ≥7.95 ± 0.13 pg, and the female nuclear DNA content is ≤7.84 ± 0.10 pg.

The mechanism of sex expression has not been investigated at the molecular level in *L. nobilis.* Sex determination by external morphology and cytogenetic studies is not user-friendly. Compared to these techniques, sex determination on a molecular basis is more effective and timesaving and provides more accurate results. Moreover, a molecular approach would help reduce the efforts of breeders and cultivators in saving field space and time. Several molecular markers have been developed and characterized to a certain extent, proving beneficial for discriminating male from female plants in several dioecious crops [111].

## 9. Biological Activity of *L. nobilis* Essential Oil

### 9.1. Antioxidant Activity

The antioxidant activity of *L. nobilis* EO has been reviewed in scientific literature, considering the importance of this aromatic plant in food formulations and traditional medicine [4,13,112,113].

In different studies, the antioxidant activity of the *L. nobilis* EO from a specific geographical area was compared to other origins. In this respect, Riabov et al. [85] evaluated two commercial EOs: one from Serbia (SRBL) and another one from Russia (RFBL). The authors did not clearly state whether the EOs were obtained from leaves. SRBL had higher total reduction capacity (TRC) and DPPH values than RFBL, and the values observed for TRC (14.59 ± 0.58 for RFBL and 17.33 ± 0.66 for SRBL) were in line with those found by Dammak et al. [114] in Tunisia and Olmedo et al. [115] in Argentina.

Sahin Basak and Candan [116] compared the in-vitro antioxidant activity of the EO, obtained from *L. nobilis* leaves collected in Iran, with that of the three major components: 1,8-cineole (68.82%), 1-(S)-α-pinene (6.94%), and R-(+)–limonene (3.04%). As generally confirmed by the scientific literature on the antioxidant and antimicrobial activity of EOs, the antioxidant activity of the phytocomplex was, in all cases greater than that of the individual compounds. Among the five tests used to evaluate the antioxidant activity, only for lipid peroxidation was the inhibition greater with 1,8-cineole (68.82%) and R-(+)–limonene (3.04%) than with the EO. In another study, Politeo et al. [63] compared the antimicrobial activity of the EO extracted from leaves of *L. nobilis* collected in Croatia with the corresponding volatile aglycones. The composition of the aglycones was completely different from the EO, apart from eugenol. In vitro results obtained with two methods (DPPH and FRAP) demonstrated lower reducing power and free radical scavenging activity of the volatile aglycones than EO [117,118,119].

The antioxidant activity of the EO extracted from the leaves of *L. nobilis* grown in Hatay, Turkey, was compared to two common antioxidants found in foods: butylated hydroxytoluene (BHT) and ascorbic acid [120]. Also, in this case, the major compound identified was 1,8-cineole (51.8%). In this study, the EO was found to have a lower reducing activity compared to the synthetic antioxidants. In contrast, the EO extracted from the floral buds of the same plant, grown in Tunisia, whose main components were *α*-terpinyl acetate (28.43%) and methyl eugenol (19.57%), was found to exert a higher antioxidant activity than BHT.

Different applications have been proposed in the scientific literature for the antioxidant activity of *L. nobilis* EO. Taoudiat et al. [121] added *L. nobilis* EO (0.01 *v*/*v*) to Algerian extra virgin olive oil stored in different packaging materials (PET, brown and transparent glass), exposed to fluorescent light for 90 days at 25  ±  2 °C. Bioprotection of extra virgin olive oil in brown glass samples was demonstrated by the highest amounts of chlorophyll and carotenoids. However, the treatment was not sufficient to avoid oxidation until the end of the experimental period.

Various extraction methods have been suggested for increasing the antioxidant activity of *L. nobilis* EO, including supercritical CO_2_ [122], microwave [96], and enzymes [94]. However, these extraction methods are not in agreement with the international definition of EO, given by ISO 9235: 2013-2.11 (ISO, 2021). Apart from the scientific literature on *L. nobilis* EO, other research has been carried out on the in-vitro antioxidant activity of *L. nobilis* leaf infusions [123], berry extracts [124], leaf extracts [125], and seed oil extracts [126], but also in vivo in the mice liver and blood hemolysate [127]. Due to differences in extraction methods [124] and extract composition, comparing these data is extremely difficult. Moreover, as evidenced by Bozan et al. Karakaplan [124], no correlation was found between total phenolics and antioxidant activity. In some studies, the extract’s antioxidant activity was higher than the EO; in particular, in a study carried out by Ramos et al. [80] on *L. nobilis* samples grown in Portugal, the highest antioxidant activity was observed in hot/cold water extracts.

### 9.2. Antimicrobial Activity

Since ancient times, *L. nobilis* has been an important ingredient in traditional medicine for the treatment of different infectious diseases. Consequently, the antimicrobial activity of *L. nobilis* EO is well known and documented in the scientific literature [4,112]. However, most of the published data were obtained in vitro, in some cases using diffusion methods [62], which cannot be considered suitable to obtain quantitative and reliable information; in fact, agar diffusion should be generally used as a preliminary check for antimicrobial activity prior to more detailed studies [128].

As for the antioxidant activity, many studies on the antimicrobial activity of *L. nobilis* EO were focused on screening the antimicrobial potential of an EO obtained in a specific geographical area. For instance, Derwich et al. [79] screened the antimicrobial activity of the EO obtained from *L. nobilis* grown in Morocco and found minimum inhibitory concentration (MIC) values ranging from 0.01 to 1 mg/mL for all the bacterial strains tested, in agreement with values obtained by Nabila et al. [129] in Algeria. Caputo et al. [130] evaluated the antimicrobial activity of the EO obtained from leaves collected in Southern Italy, containing 1,8-cineole (31.9%), sabinene (12.2%), and linalool (10.2%) as major components. For all the tested strains with the exception of *Bacillus cereus*, MIC values confirmed that the EO was more effective than 1,8-cineole, whereas for *B. cereus* the MIC was the same for the EO and its main component.

A considerable amount of research was conducted on the antifungal properties of *L. nobilis*, which could be exploited both in agriculture and in the food industry. Bayar et al. [131] investigated the activity of *L. nobilis* EO against important plant pathogens. The EO, used in vitro at 10 µL/petri dish, totally inhibited the growth of *Alternaria solani*, which causes early blight in tomatoes and potatoes, and *Sclerotinia sclerotiorum*, the agent of the white mold. Xu et al. [132] evaluated the antifungal activity of a Chinese *L. nobilis* EO in vivo against *Alternaria alternata* in cherry tomatoes, obtaining an inhibition ratio of 33.9%. This result is in agreement with inhibition values obtained in vitro (36% inhibition with 1 mg/mL EO) against *Alternaria alternata* isolated from cucurbits [133]. *L. nobilis* EO, due to its wide antifungal activity, has been proposed as a bio-preservative for cereals. In particular, Belasli et al. [134] obtained an 86% decrease in aflatoxin B1 production with 1.5 mg/mL of EO in vitro and protection between 51.5 and 76.7% against *Aspergillus flavus* during 6-month storage in fumigated wheat grains.

Few studies have been aimed at investigating the mechanism of action of the antifungal activity of *L. nobilis* EO. Rangel Peixoto et al. [135] evaluated the activity against *Candida* spp. strains, to develop formulations for the treatment of oral candidiasis. The sorbitol assay (microdilution in the presence of sorbitol) was carried out to determine the mode of action on the fungal cell wall, while the ergosterol assay (microdilution in the presence of exogenous ergosterol) was conducted to study the effect on the yeast cell membrane. According to the results, the activity of *L. nobilis* EO against *Candida* spp. is likely to be related both to cell wall biosynthesis and ionic permeability of the yeast membrane. Another clinical application of the antifungal activity of *L. nobilis* EO was proposed for the treatment of *Cryptococcus neoformans* [135].

The flowers EO, obtained from *L. nobilis* samples grown in Morocco, showed interesting MIC values, ranging from 0.05 to 0.46 mg/mL, against seven fungal strains, but the study did not state the origin of the strains, in particular, if they were clinical, environmental or food isolates [89]. In this study, the significant antifungal and antioxidant activity were attributed to 1,8-cineole as a major component of the EO (45.01%). Another study carried out in Morocco [136] aimed to evaluate possible synergistic effects between *L. nobilis* and *Myrtus communis* EOs, and different physical food preservation treatments. These authors observed a synergistic effect on foodborne pathogens by using both EOs at 0.2 µL/mL, together with mild heat processing (54 °C for 10 min) or high hydrostatic pressure (175–400 MPa for 20 min).

Other food applications for the antimicrobial activity of *L. nobilis* EO were suggested in the literature. In particular, da Silveira et al. [137] used this EO at 0.05 g/100 g or 0.1 g/100 g for shelf life extension and pathogen control in fresh Tuscan sausage stored at 7 °C. The treatment decreased total coliforms (2.8 Log CFU/g) and allowed an extension in the shelf life of two days. Although sensory characteristics of the products were affected by the presence of the EO, the product was considered acceptable by a panel of 100 non-trained consumers.

As already mentioned, for the antioxidant activity and antimicrobial activity, the possibility of increasing the effect by obtaining the EO through supercritical CO_2_ was studied [138]. In this case, the study aimed to control post-harvest spoilage fungi, namely *Botrytis cinerea*, *Monilinia laxa,* and *Penicillium digitatum*. In vivo tests were carried out on different fruit species, and the results were indeed promising for the protection of kiwifruit and peach against *B. cynerea* and *M. laxa*, respectively.

Finally, in addition to EOs, hydrosols have promising perspectives for food applications due to their interesting content of bioactive compounds, coupled with water solubility that facilitates their use in several food environments. In the review by D’Amato [139], data were gathered on the effects of the hydrosol of *L. nobilis* on fresh-cut vegetables and fruits.

### 9.3. Effect on Arthropodos

Many plant secondary metabolites have physiological and behavioral effects on arthropods. These effects may include toxicity, repellency, attraction, and anti-feeding effects on insect pests [140,141]. Essential oils also contain compounds that affect egg-laying, repellency, sterilization, anti-feeding, and insect toxicity [142].

Essential oils from *L. nobilis* have considerable repellent effects on *Ephestia kuehniella* Zeller (Lep: Pyralidae); at the highest concentration (2.00 µL·L^−1^ air), repellency rates were reached 84.2%, and even at (0.50 µL·L^−1^ air) low concentrations it reaches 20.4% repellency [143].

*L. nobilis* essential oils from Tunisia, Algeria, and Morocco were found repellant and toxic against two other major stored product pests, the stored grain borer *Rhyzopertha dominica* and rust-red flour beetle *Tribolium castaneum*. Results showed that *L. nobilis* essential oils repellent and fumigant toxicities depended on insect species and oil origin. In filter paper tests, RD50 values ranged from 0.013 μL/cm^2^, to 0.036 μL/cm^2^ for *R. dominica* and from 0.045 μL/cm^2^ to 0.139 μL/cm^2^ for *T. castaneum*. Moreover, fumigant activity tests showed that both *R. dominica* and *T. castaneum* were more susceptible to *L. nobilis* essential oil from Morocco than that from Algeria or Tunisia [59].

*L. nobilis* extracts were also tested against the main honeybee pests. The extract of ethanol showed minimal inhibitory concentration (MIC) values of 208 to 416 μg/mL and showed significant antiparasitic activity on *Varroa destructor*, killing 50% of mites after 30 s of exposure [144]. When tested against the aphid *Aphis gossypii* Glover (Hem: Aphididae), The LC50 value of the essential oil was 3.16 μL·L^−1^ air [145]. Laurel essential oils also exhibited an interesting fumigant larvicidal activity against moth larvae. When tested against the date moth *Ectomyelois ceratoniae* (Pyralidae), respective LC50 and TL50 values were 750.4 μL·L^−1^ air and 33.8 days [146].

However, not all arthropods feeding on the bay are affected; for example, *Antocoris nemoralis,* the natural enemy of the Laural Psylla *Trioza alacris* (Flor), is not affected negatively by the Laural and can be found on the bay leaves feeding on the psylla [147] and has a high fitness feeding on Laural pollen and sap (Shaltiel unpublished data).

## 10. Conclusions

*L. nobilis,* an evergreen shrub widespread in Mediterranean countries is primarily used as a culinary herb. 1,8-cineole is reported as a significant volatile in the EO along with sabinene, *α*-terpinyl acetate, linalool, *α*-pinene, *α*-terpineol, and methyl eugenol, with varying concentrations from different locations. Various EO extraction methods have been developed to decrease the environmental impact by using less or no solvent and less energy, while maintaining a good yield in volatiles. It has been increasingly acknowledged over the years as it provides an essential contribution to the food and pharmaceutical industries and cultural integrity. In particular, a considerable amount of research has been carried out on the antioxidant and antimicrobial properties of the EO, but still important issues need to be clarified, for example, modes of action, toxicity, cost reduction, and mode of application.

Despite the importance of this plant species, little conservation efforts and a lack of knowledge about the diversity and characterization of genetic resources are probably the major causes of the lack of breeding programs. In this regard, it is fundamental to characterize the available germplasm collections in order to identify and classify accessions and generate a catalog of information for germplasm management and further breeding programs. The characterization of germplasms based on molecular methods has fomented a revolution in speed and quality. In recent years, molecular tools have been used to elucidate some aspects of genetic diversity in aromatic plants, the genetic relationships between different cultivars, and molecular marker associations to plant chemical composition. However, understanding the molecular basis of flavor and aroma production is not an easy task to accomplish. Several aspects of plant secondary metabolism, and in particular volatile production in *L. nobilis*, are still unknown. Finally, the routes from genomics to proteomics are not illustrated. Addressing these questions requires concentrated breeding efforts and multidisciplinary approaches, including genomics, transcriptomics, proteomics, and bioinformatics, which might open up new perspectives in future *L. nobilis* improvement programs.

## Figures and Tables

**Figure 1 plants-11-01209-f001:**
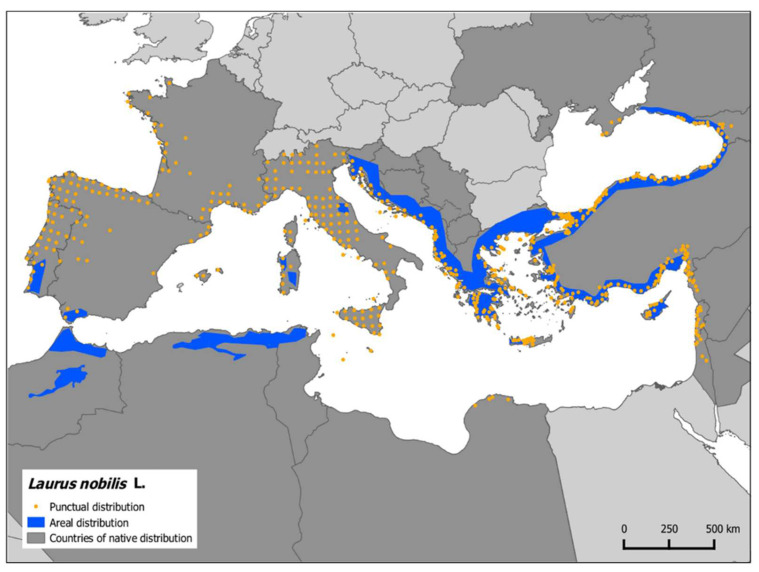
Distribution of *L. nobilis* L. Source-FAO.

**Figure 2 plants-11-01209-f002:**
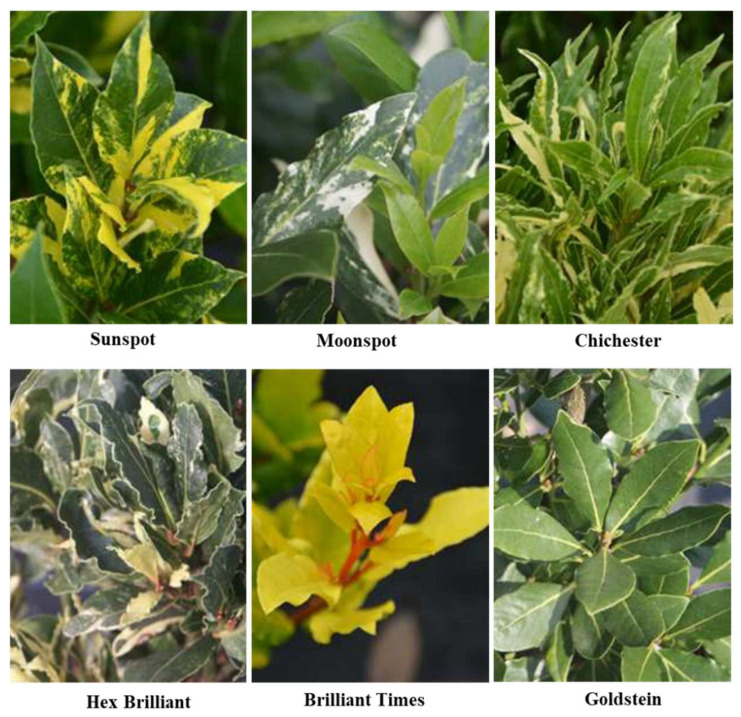
Variation in leaf morphology of some of the *L. nobilis* L. cultivars. Source: https://sierplant.be/soorten/ (accessed on 3 April 2022).

**Figure 3 plants-11-01209-f003:**
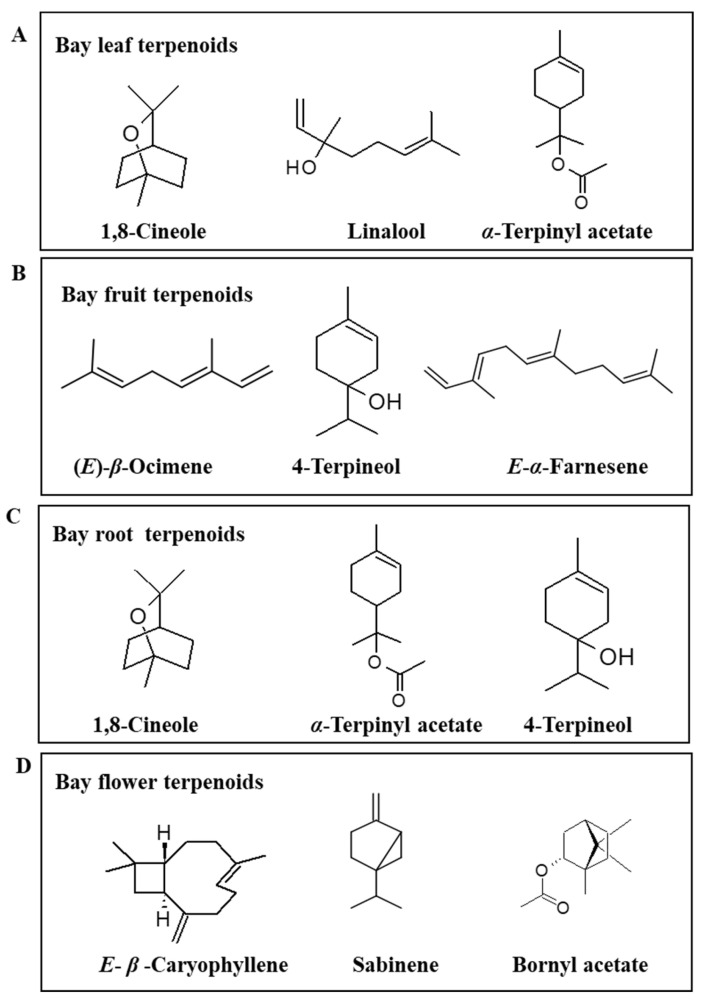
Representative terpenoids biosynthesized by bay (*Laurus nobilis* L.). (**A**) Bay leaf terpenoids; (**B**) Bay fruit erpenoids. (**C**) Bayroot erpenoid; (**D**) Bayflower erpenoid.

**Table 1 plants-11-01209-t001:** Salient features of selected bay cultivars.

Cultivar Name	Main Features	Common Name
*Angustifolia*	Blades much narrower, usually lanceolate to oblanceolate. Leaves are a lighter or brighter green than typical species.	Willow-leaved laural
*Aurea/Aureum*	Golden yellow foliage, brightest in winter and spring, and dense clusters of small, greenish-yellow male flowers in spring	Golden bay
*Baccalia/Bacalia*	In old literature, *Bacalia* was mentioned as the commonest laurel and the one that bears the greatest number of berries.	
*Bay junior*	Compact and slower growth habitat grows more slowly with a smaller leaf but produces very full, dense foliage-medium-wide elliptic leaves.	
*Borms*	The cold-resistant cultivar produces fewer or no flowers	
*Borziana var. borzianaBég*	Elongated, narrower leaf with a thinner blade; fruits more spherical large to ca. 1/2 inch.	Borzis bay
*Brilliant times*	Stem is often reddish in new shoots, contrasting and showy at times. Leaves are bright yellow, faintly tinged red on margins at young later yellowish-green.	
*Celtic queen*	Fast-growing—can grow to 8 feet tall in 3 years and is hardy tolerating as low as −4 F, deep green foliage	
*Chichester*	Mottled and sectored leaf with narrow elliptic to linear-elliptic blade much as ‘Angustifolia’ with chimera	
*Crispa/Undulata*	Leaf margin distinctly undulate, egg-shaped fruits, low shrub seldom higher than 4 to 6 feet.	Wavy-leaved bay
*Cylindrocarpa*	Leaves are fairly wide and ovate; cylindrical fruits.	Cylinder-fruited bay
*Cyprian*	Shorter in height, darker, blackish-green leaves and their margins are crisped and imbricated.	
*Delphic*	Leaves are uniformly green, and fruits are green-tinged red with large sizes.	
*Emerald Wave@Monem*	Wavy-edged, emerald green leaves; slender, upright form grows slowly and gracefully into a small pyramidal tree, ideal for topiary.	
*Eriobotryfolia*	Very large, wide, toothed leaves resemble Eriobotrya.	Loquat-leaved bay
*Feys*	Cold hardy cultivar.	
*Flore Pleno*	A double-flowered version of the broad-leaved Dutch myrtle.	Double-flowered bay
*Glauca*	More glaucous or bluish (waxy) leaves.	Glaucous-leaved bay
*Goldstein*	Slightly undulate classic bay leaf with white midrib.	
*Grandiflora*	Larger fruit diameter than average.	
*Hex brilliant*	Leaves irregularly margined white to cream and with a thin margin, irregularly twisted and very asymmetrical.	
*Holy land*	Origin–Israel, leaves are slightly wavy on the edges.	
*Latifolia*	Leaves are broad and smooth, less hardy than most other varieties, native to Spain, Italy, and Asia.	Broad-leaved bay
*Ligustrifolia*	Privet-shaped leaves.	Privet-leaved bay
*Little Ragu^®^ ‘MonRik’*	A more compact sweet bay with highly aromatic, deep green foliage that emerges on unique red-tinged stems. Fabulous fragrance.	
*Macrocarpa*	Much larger fruit diameter.	Large-fruited bay
*Macroclada*	Tall plants, leaves are rounded apex, tapering to base, stout while fruits are globular, large, ca. 1/2 inch.	Large bay
*Microcarpa*	Small fruit diameter.	
*Moonspot*	Leaves mottled and sectored white to 35%.	
*Multiflora*	Flowers in multiples or numerous.	Many-flowered bay
*Mustax*	Very large blade and the blades are more flaccid. Leaves are a whitish hue. It is named for its use in cakes called mustaceum.	Mustaceum
*Nancy howard*	Roots are hardy to Philadelphia, Pennsylvania.	Nancy Howard bay
*Olivaeformis*	Fruits are more olive-shaped.	Olive-fruited bay
*Ovalifolia*	Distinctly oval-shaped leaf.	Oval-leaved bay
*Pallida*	Leaves are yellow-green; fruits have reddish, shrubby growth.	
*Parvifolia*	Small leaves; French origin Caribbean.	Small-leaved bay
*Pedunculata*	Flowers are more pedunculate.	Pedunculate-flowered bay
*Popeye*	Narrowly pyramidal tree with shorter internodes and compact for small spaces.	
*Pride of Provence*	Compact growing bay tree with lush dark green shiny foliage; shorter internodes, requiring little or no trimming.	Hedge bay laurel
*Rotundifolia*	Leaves are wide, mostly round.	Round-leaved bay
*Rubrinervis*	Red-veined leaves.	Red-veined bay
*Salicifolia*	Narrow lance-shaped leaves are not as thick as the normal variety, have a lighter green color, shrubby growth (1.8 to 2.4 m); Mostly confused in nursery trade with ‘Angustifolia.’	Willow-leaf bay
*Saso’s dwarf*	Leaves are thick, dark green; shrubby growth	
*Sphaerocarpa*	Leaves are small, polished, spherical fruits.	Ball-fruited bay
*Sunspot*	Gold-variegated foliage.	Sunspot sweet bay
*Variegata*		gold-striped bay

Along with these cultivars, many cultivars and/or genotypes have been enlisted in the literature, but detailed information is not available for them.

**Table 2 plants-11-01209-t002:** Variation in the major volatile composition from essential oils (%) of bay plant parts from different geographic regions.

Leaves											
Location	1,8–Cineole	Sabinene	*α*-Terpinyl Acetate	*α*-Pinene	Linalool	Methyl Eugenol	*β*-Pinene	Eugenol	Camphene	*α*-Terpineol	Reference
Albania	26.70	11.80	12.00	2.20	18.50	2.50	2.40	6.50	0.20	1.20	[57]
Algeria	17.6–44.13	2.20–9.60	7.90–17.33	0.90–9.20	4.18–12.57	5.10–11.0	0.80–3.80	1.20–3.60	0.20–8.91	2.58–7.6	[58,59]
Argentina	37.3–43.8	8.4	7.9–10.6	4.8	12.7–19.4	3.0–3.6	3.5–20.1	0.7	0.4	2.2–2.8	[60]
Brazil	26.9–37.3	10–4−13.7	15.3–17.4	8.9–10.9	0.4–0.6	0.71–0.72	-	0.39	0.44–0.51	0.04	[61]
Bulgaria	41.0	8.8	14.4	2.56	4.92	6.0	2.45	1.47	0.18	3.11	[62]
China	25.5–43.0	2.1–6.8	9.5–18.0	1.0–5.2	4.4–22.7	1.7–8.9	1.0–4.4	1.4–2.4	0.2	1.5–2.5	
Colombia	22.0	1.5	11.1	2.9	16.4	2.9	6.1	2.0	0.1	4.9	
Croatia	45.5	5.7	9.1	2.1	8.5	10.0	--	2.5	--	1.5	[63]
France	39.1	4.4	18.2	2.2	10.0	11.8	1.7	--	0.2	1.3	[64]
Germany	23.3	5.3	9.8	4.6	2.0	3.5	4.0	2.9	0.7	3.3	[65]
Greece	49.60	7.80	5.25	5.96	1.50	2.10	5.12	5.60	1.21	1.90	[66,67]
India	0.27–3.31	0.22	--	1.37–7.68	29.08–50.68	1.00	0.71–1.52	63.57	0.26–1.14	0.39–0.60	[68]
Iran	25.7–61.0	2.33–8.7	6.14–15.14	2.35–6.59	1.40–3.96	3.08–5.18	1.40–4.6	0.4–2.88	0.27–10.22	0.69–3.88	[69,70,71,72,73]
Israel *	1698.2–2549.7	168.8–277.1	255.6–482.3	58.6–77.6	106.7–151.5	36.8–40.2	37.2–69.2	7.5–11.8	4.2–4.5	184.5–233.1	
Italy	22.84–35.70	4.30–6.50	4.43–14.23	2.6–5.72	7.08–19.47	2.52–16.22	2.40–3.46	1.6–5.97	0.14–0.30	2.42–6.44	[74,75,76]
Jordan	36.80–40.91	3.10–6.92	5.86–14.6	4.60–5.82	1.29–2.60	1.62–4.20	3.60–4.55	0.92	0.50–0.58	--	[17,77]
Morocco	35.62–58.88	0.42–6.13	0.45–8.96	3.72–4.58	1.98–9.45	1.70–3.93	1.92–3.14	0.56–1.97	0.5–4.87	1.56–5.83	[78,79]
Lebanon	57.05–65.99	4.06–9.74	--	2.14–6.03	0.51–0.75	--	2.51–4.17	--	0.14–0.17	2.90–3.64	
Nepal	1.64–26.64	0.26–0.43	--	2.05–6.70	28.97–72.67	--	1.17–3.03	0.21–0.23	0.49–1.68	0.37–3.05	[52]
Portugal	--	4.00	10.20	2.3	8.4	5.4	1.8	1.2	0.1	2.7	[80]
Turkey	46.61–72.09	4.44–14.05	4.04–25.70	2.19–6.11	0.37–1.9	0.41–3.39	2.58–3.91	0.40–1.55	0.19–0.67	0.95–6.83	[3,14]
Spain	33.28–43.56	2.48–5.80	6.0–11.75	0.85–11.6	6.50–26.70	3.12–4.64	3.47–4.90	1.23–3.00	0.7–1.06	0.9–4.95	[57,81]
Serbia	15.54–41.86	0.55–9.12	5.49–24.74	0.12–7.20	1.81–16.00	5.32–8.67	0.16–5.23	1.07–6.14	0.06–0.83	1.65–4.28	[82,83,84,85]
Syria	58.66–73.70	3.56–8.89	--	2.62–3.85	0.32–0.97	--	2.66–3.25	--	0.05–0.27	1.12–3.50	[17]
Fruit											
Location	1,8–Cineole	Sabinene	*α*–Terpinyl acetate	*α*–Pinene	(*E*)-*β*-ocimene	*α*–Phellandrene	*β*-Pinene	*β*-Elemene	Camphene	Germacrene-A	Reference
Bulgaria	33.33	6.30	10.30	11.01	0.72	5.18	0.28	7.45	4.33		[62]
Turkey	9.50–20.45	1.70–6.03	1.20–4.88	3.3–16.55	11.88–28.35	10.58–15.87	2.1–12.83	2.0–4.46	0.80–2.08	0.80–4.35	[3,12]
Jordan	29.8	4.4	1.2	10.9	3.2	9.0	8.4	6.2	1.3	0.2	[77]
Lebanon	17.64–48.01	2.93−4.49	0.87–2.07	7.69–17.96	0.57–11.82	8.29–17.07	3.91–9.51	-	1.08–2.61	-	
Iran	14.4–46.7	5.4	5.8.5	2.8–6.6	20.8–22.1	4.7	5.1–7.3	2.1–3.5			[86]
Tunisia	8.1–8.8	1.8–2.6	3.0–3.8	8.0–10.3	20.9–23.7		4.2–5.8		2.6–3.8		[87]
Israel *	800.0–1278.5	97.6–103.1		312.4–340.1	1262.9–1284.6	204.7–212.6	228.5–242.0	411.8–373.1	53.2–55.5	904.2–1012.9	
Flower											
Location	1,8–Cineole	Sabinene	*α*–Terpinyl acetate	*α*–Pinene	(*E*)-*β*-ocimene	Linalool	*β*-Pinene	*β*-Elemene	Bornyl acetate	*E-β*-Caryophyllene	Reference
Turkey	8.8	1.7	1.8	5.1	2.7	-	3.7	5.4	2.1	5.1	[12]
Italy	7.9–42.8	0.5–6.0	0.3–12.0	0.9–3.8	0.1–65.3	0.6–14.4	0.4–2.9	0.1–5.3	0.1–0.6	0.1–15.4	[88]
Morocco	45.01	3.01	0.1	3.04	-	1.04	3.01	-	0.25	0.42	[89]
Israel *	1186.4–3526.1	328.5–344.1	1314.4	278.7–452.1	1046.	346.7	401.4	421.0–498.1	1019.7	1715.1	

* Concentration in ng g^−1^ of fresh weight.

**Table 3 plants-11-01209-t003:** Comparative studies of improved extraction methods and their effect on volatile composition in *Laurus nobilis*.

Sample Number	Extraction Method	Features/Treatment	Effect on Volatiles Composition	Reference
1	Enzyme assisted extraction	Pre-treatment with cellulase, hemicellulase, xylanase for 1 h at 40 °C	Increase in 1,8-cineole, methyl eugenol, terpinen-4-ol, *α* terpineol, and caryophyllene oxide than hydrodistillation	[94]
2	Simultaneous distillation extraction (SDE)	Extraction for 2 h in dichloromethane using microscale simultaneous distillation apparatus	More hydrocarbon monoterpenes (15.7%) as with from SPME (3.6%)	[81]
3	Solid-phase microextraction (SPME)	Adsorption in headspace using dimethylsiloxane fiber (100 µm) at 60 °C for 30 min	Oxygen terpenes (95.7%) as compared with SDE (83.4%)	[81]
		Sample 3 mg instead of 50–200 mg	Identified 98 different volatiles	[76]
4	Supercritical carbon dioxide (SFE)	Pressure, 100 bar; temperature, 40 °C; and CO_2_ flow, φ 0.3 kg/h; time, 1.4 h	Two times less monoterpene hydrocarbons and oxygenated monoterpenes (43.89%) in comparison to HD (98.4%).	[84]
		Pressure, 100/250 bar; temperature, 40/50 °C; and CO_2_ flow, φ 1.629 dm^3^/min; time, 3 h	100 bar/40 °C and 250 bar/40 °C had the highest amount of 1, 8-cineole, but at 100 bar/60 °C, *α*-terpineol acetate was dominant.	[82]
		Pressure, 90 bar; temperature, 50 °C; and CO_2_ flow, φ 1.0 kg/h.	Monoterpenes were extracted in shorter times than sesquiterpenes	[74]
		Pressure, 80 bar; temperature, 40 °C and 50 °C; and CO_2_ flow, φ 1 mL/min; time, 20 min.	A decrease in monoterpene hydrocarbons (6.8–5.1%) in comparison to hydro–(19.5%) and steam (16.5%) distilled oils	[95]
5	Solvent-Free Microwave Extraction (SFME)	Power, 850 W for 30 min	1,8-cineole, linalool, eugenol and Methyl eugenol content increased	[96]
		Power, 300, 600, and 900 W for 20 min	Oxide volatile organic compound reduced in comparison to HD. The highest 1,8-cineole (72%) content at 300 W	[96]
		Power, 622 and 249 W for 1 h	No significant differences in volatiles by SFME and HD but 55–60% time was reduced by SFME	[96]
		Power, 200, 300 W and pulsed microwaves system for 1 h	Higher amounts of oxygenated compounds and lower amounts of monoterpenes	[76]
6	Optimum ohmic heating assisted hydrodistillation (OAHD)	120 min, 8.53 V/cm, and 40 g	EO yield of OAHD was found to be higher when compared with HD methods	[97]

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
