# Peer review of "A Review of the Botany, Volatile Composition, Biochemical and Molecular Aspects, and Traditional Uses of Laurus nobilis"

_plants, 2022, doi:10.3390/plants11091209_

Round 1
Reviewer 1 Report
Dear Authors,
Your manuscript "A review of the botany, volatile composition, biochemical and molecular aspects, and traditional uses of Laurus nobilis" is interesting.
Laurus nobilis, it is a very popular plant since ancient times in traditional medicine due to their beneficial health effects. Also bay leaf is used as a flavoring agent in the kitchen of different nationalities for meat, fish, broths, and vegetables. Laurel essential oil have been recognized to possess many interesting properties that have potential applications in many areas, including agriculture, medical, food, pharmaceutical industries, etc.
All parts of the manuscript are well described and supported by the relevant tables and figures. In my opinion, about 40 sources from the last 3-5 years are not enough for a review article.
I would like to pay special attention to the authors of the "References" section. Here the sources that are noted are up to number 151, but there are 37 other sources that are not numbered by the authors. Is this an omission by the authors or a bug in the software they used?
I would like the authors to draw attention to some typographical errors in the spelling of chemical formulas such as 1.8 cineole, 1,8-cineole is correct, also when writing the dimensions should be written in the same way, for example: µl. L-1, μl/cm2 (to correct font), i.e. µL.L-1, μL/cm2 as at μg/mL.
Also in the References section there is an article that does not mention the year Belasli, A., Ben Miri, Y., Aboudaou, M., Ouahioune, L.A., Montañes, L., Ariño, A., Djenane, D. Antifungal, antitoxigenic , and antioxidant activities of the essential oil from laurel (Laurus nobilis L.): Potential use as a preservative wheat. Food Science & Nutrition 8 (9): 4717-4729.
Also in the abbreviation of essential oil (EO) is converted to OE.
Last but not least, I would like to say that once the manuscript is corrected, it would be of interest not only to readers of the Plant Journal but also to researchers in the food and pharmaceutical industries, agriculture and others.
I think that after the authors carefully correct their manuscript it can be accepted.
Author Response
Reviewer 1
Your manuscript "A review of the botany, volatile composition, biochemical and molecular aspects, and traditional uses of Laurus nobilis" is interesting.
Laurus nobilis, it is a very popular plant since ancient times in traditional medicine due to their beneficial health effects. Also bay leaf is used as a flavoring agent in the kitchen of different nationalities for meat, fish, broths, and vegetables. Laurel essential oil have been recognized to possess many interesting properties that have potential applications in many areas, including agriculture, medical, food, pharmaceutical industries, etc.
All parts of the manuscript are well described and supported by the relevant tables and figures.
In my opinion, about 40 sources from the last 3-5 years are not enough for a review article.
Response: As suggested we have added some new sources in the revised manuscript. We tried to cover all relevant sources. Please let us know if any missing.
I would like to pay special attention to the authors of the "References" section. Here the sources that are noted are up to number 151, but there are 37 other sources that are not numbered by the authors. Is this an omission by the authors or a bug in the software they used?
Response: Corrected
I would like the authors to draw attention to some typographical errors in the spelling of chemical formulas such as 1.8 cineole, 1,8-cineole is correct, also when writing the dimensions should be written in the same way, for example: µl. L-1, μl/cm2 (to correct font), i.e. µL.L-1, μL/cm2 as at μg/mL.
Response: Corrected
Also in the References section there is an article that does not mention the year Belasli, A., Ben Miri, Y., Aboudaou, M., Ouahioune, L.A., Montañes, L., Ariño, A., Djenane, D. Antifungal, antitoxigenic , and antioxidant activities of the essential oil from laurel (Laurus nobilis L.): Potential use as a preservative wheat. Food Science & Nutrition 8 (9): 4717-4729.
Response: Corrected
Also in the abbreviation of essential oil (EO) is converted to OE.
Response: Corrected
Last but not least, I would like to say that once the manuscript is corrected, it would be of interest not only to readers of the Plant Journal but also to researchers in the food and pharmaceutical industries, agriculture and others.I think that after the authors carefully correct their manuscript it can be accepted.
Reviewer 2 Report
A very interesting work, very well planned and presented. I present a few remarks below.
Line 44. Almost 97% of the world total production is produced by Turkey leaves – Please, improve the style, production comes from Turkey or they are leaves from Turkey, or Turkish leaves
Line 97. 3. Origin, Domestication, and Spread. There is no information in this chapter about the laurel forests (Laurisilva) of the Mediterranean coast, which are also found in Macaronesia, as a Tertiary relic.
Line 124. It can grow in the garden and be as a hedge , Suggested: It can grow in the garden, also as a hedge.
Line 282. Table 2. To wonder if the letters A, B, and C are needed for the Leaf, Fruit, and Flower chakatreistic? You can omit them, leaving the names of the organs of the plant in the header, without any letters.
Line 661. The conclusions should summarize the entirety of the topic, but it seems that the authors paid most attention to the molecular aspects (over half of the chapter). Perhaps it would be worth emphasizing more other aspects of the topic of the article (botany, volatile composition, biochemical aspects, and traditional uses ), and reducing the genetic part.
Author Response
Reviewer 2
A very interesting work, very well planned and presented. I present a few remarks below.
Line 44. Almost 97% of the world total production is produced by Turkey leaves – Please, improve the style, production comes from Turkey or they are leaves from Turkey, or Turkish leaves
Response: Done
Line 97. 3. Origin, Domestication, and Spread. There is no information in this chapter about the laurel forests (Laurisilva) of the Mediterranean coast, which are also found in Macaronesia, as a Tertiary relic.
Response: Thank you for pointing it. We have added very brief information on it.
Line 124. It can grow in the garden and be as a hedge , Suggested: It can grow in the garden, also as a hedge.
Response: Corrected
Line 282. Table 2. To wonder if the letters A, B, and C are needed for the Leaf, Fruit, and Flower chakatreistic? You can omit them, leaving the names of the organs of the plant in the header, without any letters.
Response: Done
Line 661. The conclusions should summarize the entirety of the topic, but it seems that the authors paid most attention to the molecular aspects (over half of the chapter). Perhaps it would be worth emphasizing more other aspects of the topic of the article (botany, volatile composition, biochemical aspects, and traditional uses ), and reducing the genetic part.
Response: As per suggestion conclusion section is modified in revised manuscript.
Reviewer 3 Report
This review, titled "A review of the botany, volatile composition, biochemical and molecular aspects, and traditional uses of Laurus nobilis," attempted to compile and review all available literature on the phytochemical composition, extraction procedures, traditional medicinal uses, biological activities, morphology, and habitat of L. nobilis. The review is interesting and eventually contains relevant information for readers. However, there are many points to be addressed by the authors. The authors simply reviewed the literature without attempting to discuss the findings, which is regarded as the review's major weakness. Discuss the differences in chemical composition as well as biological properties in greater depth. The authors used long and sometimes illegible sentences, which could lead to reader confusion. The language style must be revised, and there are numerous editing errors.
Line 42: Turkey is the major producer of L. nobilis and exports it to 64 countries, including 42 Vietnam, Germany, Poland, the U.S.A., Hong Kong, Japan, Brazil, the People's Republic 43 of China, and the Netherlands [4,8]. There is no need to include a list of nations that import L. nobilis from Turkey. It makes no difference to the reader.
The authors should List the Essential oil composition of L. nobilis
Converting Israel's EO composition to percentages is necessary.
Line 315: change δ-cardinene > δ-cadinene
Line 392: a-terpinyl acetate> a-terpinyl acetate
Line 398: Along with the disappearance of some volatiles, these authors also observed specific volatiles in air and infrared (IR) (65°C) drying: tricyclene, trans-2-hexenal, trans-2-hexenol, a-terpinene, terpinolene, camphor, linalyl acetate, allylanisole, myrtenyl acetate, a-terpineol, borneol, valencene, geranyl acetate, myrtenol, nonadecane, and spathulenol. However, these compounds were absent in the fresh leaves extract.
a-terpinene> a-terpinene
a-terpineol> a-terpineol
Is there any explanation for this odd observation? The majority of the above-mentioned monoterpenes will be lost if the leaves are dried at 65 oC.
Line 534-541; rephrase the paragraph: Interestingly, Yilmaz et al. [122] compared the antioxidant activity of the EO extracted from leaves of L. nobilis grown in……
Line 641: L. nobilis essential oils from ………??were found repellent and toxic
Line 645: between 0.013 μl/cm2, to 0.036 645 μl/cm2 for R. dominica >
Ranged from 0.013-0.036 645 μl/cm2 for R. dominica
and from 0.045-0.139 μl/cm2 for T. castaneum. M
Line 649: rephrase the following sentence: The ethanol extract showed minimal inhibitory concentration (MIC) values of 208 to 416 μg/mL, showed a significant antiparasitic activity on Varroa destructor, killing 50 % of mites 24 h after a 30-s exposure [148]
Author Response
Reviewer 3
This review, titled "A review of the botany, volatile composition, biochemical and molecular aspects, and traditional uses of Laurus nobilis," attempted to compile and review all available literature on the phytochemical composition, extraction procedures, traditional medicinal uses, biological activities, morphology, and habitat of L. nobilis. The review is interesting and eventually contains relevant information for readers. However, there are many points to be addressed by the authors. The authors simply reviewed the literature without attempting to discuss the findings, which is regarded as the review's major weakness. Discuss the differences in chemical composition as well as biological properties in greater depth. The authors used long and sometimes illegible sentences, which could lead to reader confusion. The language style must be revised, and there are numerous editing errors.
Response: We agree with the reviewer, and we made the necessary changes and revised the language style too.
Line 42: Turkey is the major producer of L. nobilis and exports it to 64 countries, including 42 Vietnam, Germany, Poland, the U.S.A., Hong Kong, Japan, Brazil, the People's Republic 43 of China, and the Netherlands [4,8]. There is no need to include a list of nations that import L. nobilis from Turkey. It makes no difference to the reader.
Response: Corrected
The authors should List the Essential oil composition of L. nobilis
Response: Table 2 summarizes the major volatiles composition variation from essential oils (%) of bay plant parts and different geographic regions.
Converting Israel's EO composition to percentages is necessary.
Response: unfortunately, because the original data was giving in “ng g-1 of fresh weight,” we wont be able to modify it. The conversation from ng g-1 of fresh weight to percentages required information lacking in the original paper.
Line 315: change δ-cardinene > δ-cadinene
Response:Done
Line 392: a-terpinyl acetate> α-terpinyl acetate
Response: Done
Line 398: Along with the disappearance of some volatiles, these authors also observed specific volatiles in air and infrared (IR) (65°C) drying: tricyclene, trans-2-hexenal, trans-2-hexenol, a-terpinene, terpinolene, camphor, linalyl acetate, allylanisole, myrtenyl acetate, a-terpineol, borneol, valencene, geranyl acetate, myrtenol, nonadecane, and spathulenol. However, these compounds were absent in the fresh leaves extract.
a-terpinene> α-terpinene
Response:Done
a-terpineol> α-terpineol
Response:Done
Is there any explanation for this odd observation? The majority of the above-mentioned monoterpenes will be lost if the leaves are dried at 65 oC.
Response: Due to drying, moisture moves by diffusion to the leaf surfaces and drags EO with it and these monoterpenes might have more affinity to the water fraction contained in bay leaves and thereby, they were lost with water during drying process.
As suggestion it is explained in revised manuscript.
Line 534-541; rephrase the paragraph: Interestingly, Yilmaz et al. [122] compared the antioxidant activity of the EO extracted from leaves of L. nobilis grown in……
Response: Corrected, and the sentence read as follows:” The antioxidant activity of the EO extracted from the leaves of L. nobilis grown in Hatay, Turkey, was compared to two common antioxidants found in foods: butylated hydroxytoluene (BHT) and ascorbic acid.”
Line 641: L. nobilis essential oils from ………??were found repellent and toxic
Response: Corrected, and the sentence read as follows:” L. nobilis essential oils from Tunisia, Algeria, and Morocco were found repellant and toxic against two other major stored product pests, the stored grain borer Rhyzopertha dominica and rust-red flour beetle Tribolium castaneum Tribolium castaneum.”
Line 645: between 0.013 μl/cm2, to 0.036 645 μl/cm2 for R. dominica >
Ranged from 0.013-0.036 645 μl/cm2 for R. dominica and from 0.045-0.139 μl/cm2 for T. castaneum. M
Response: Corrected
Line 649: rephrase the following sentence: The ethanol extract showed minimal inhibitory concentration (MIC) values of 208 to 416 μg/mL, showed a significant antiparasitic activity on Varroa destructor, killing 50 % of mites 24 h after a 30-s exposure [148].
Response: Corrected, and the sentence read as follows:” The extract of ethanol showed minimal inhibitory concentrations (MIC) values of 208 to 416 μg/mL and showed significant antiparasitic activity on Varroa destructor, killing 50 % of mites after 30 seconds of exposure.”
Reviewer 4 Report
Authors of the reviewed article described different issues related to plant Laurus nobilis L. Authors provide readers with useful information concerning cultivation of L. nobilis, its geographical distribution, EO composition, as well as antioxidant and antimicrobial activity. The manuscript is well structured, with minor linguistic and punctuation that should be corrected before publication (for example line 46, 649 – lack of dot; line 197, 205 – L. nobilis should be italic; line 202, 226, 333, 634, 639 - double space; 475- lack of space, line 645-646 - superscript in the units).
Author Response
Reviewer 4
Authors of the reviewed article described different issues related to plant Laurus nobilis L. Authors provide readers with useful information concerning cultivation of L. nobilis, its geographical distribution, EO composition, as well as antioxidant and antimicrobial activity.
The manuscript is well structured, with minor linguistic and punctuation that should be corrected before publication (for example line 46, 649 – lack of dot; line 197, 205 – L. nobilis should be italic; line 202, 226, 333, 634, 639 - double space; 475- lack of space, line 645-646 - superscript in the units).
Response: Corrected
Round 2
Reviewer 1 Report
Dear Authors,
I agree with the corrections you have made to your manuscript. I think the manuscript may be accepted for publication.